# A Live Attenuated COVID-19 Candidate Vaccine for Children: Protection against SARS-CoV-2 Challenge in Hamsters

**DOI:** 10.3390/vaccines11020255

**Published:** 2023-01-24

**Authors:** Rajeev Mehla, Prasad Kokate, Sarika R. Bhosale, Vivek Vaidya, Shridhar Narayanan, Radha. K. Shandil, Mayas Singh, Gudepalya R. Rudramurthy, Chakenahalli N. Naveenkumar, Kumaraswamy Bharathkumar, Rob Coleman, Steffen Mueller, Rajeev M. Dhere, Leena R. Yeolekar

**Affiliations:** 1Serum Institute of India Pvt. Ltd., Pune 411028, Maharashtra, India; 2Foundation for Neglected Disease Research, Bengaluru 561203, Karnataka, India; 3Codagenix, Inc., Farmingdale, New York, NY 11735, USA

**Keywords:** COVID-19 in children, vaccine efficacy, SARS-CoV-2, hamster challenge study, measles, rubella, coronavirus combination vaccine, live-attenuated, codon de-optimized vaccine

## Abstract

Children are at risk of infection from severe acute respiratory syndrome coronavirus-2 virus (SARS-CoV-2) resulting in coronavirus disease (COVID-19) and its more severe forms. New-born infants are expected to receive short-term protection from passively transferred maternal antibodies from their mothers who are immunized with first-generation COVID-19 vaccines. Passively transferred antibodies are expected to wane within first 6 months of infant’s life, leaving them vulnerable to COVID-19. Live attenuated vaccines, unlike inactivated or viral-protein-based vaccines, offer broader immune engagement. Given effectiveness of live attenuated vaccines in controlling infectious diseases such as mumps, measles and rubella, we undertook development of a live attenuated COVID-19 vaccine with an aim to vaccinate children beyond 6 months of age. An attenuated vaccine candidate (dCoV), engineered to express sub-optimal codons and deleted polybasic furin cleavage sites in the spike protein of the SARS-CoV-2 WA/1 strain, was developed and tested in hamsters. Hamsters immunized with dCoV via intranasal or intramuscular routes induced high levels of neutralizing antibodies and exhibited complete protection against the SARS-CoV-2 wild-type isolates, i.e., the Wuhan-like (USA-WA1/2020) and Delta variants (B.1.617.2) in a challenge study. In addition, the dCoV formulated with the marketed measles–rubella (MR) vaccine, designated as MR-dCoV, administered to hamsters via intramuscular route, also protected against both SARS-CoV-2 challenges, and dCoV did not interfere with the MR vaccine-mediated immune response. The safety and efficacy of the dCoV and the MR-dCoV against both variants of SARS-CoV-2 opens the possibility of early immunization in children without an additional injection.

## 1. Introduction

COVID-19 is a disease caused by severe acute respiratory syndrome coronavirus-2 (SARS-CoV-2). As of December 2022, 641 million COVID-19 cases with nearly 6.6 million deaths (~1% mortality) have been reported worldwide [1]. COVID-19 symptoms include moderate to severe lung infection and pneumonia. COVID-19 incidences were reported in children following relaxation in social and public health measures and emergence of SARS-CoV-2 variants [2,3,4]. Of utmost concern is children having COVID-19 associated complications such as neurological complications, multisystem inflammatory syndrome in children (MIS-C) and long COVID and requiring intensive care [5,6,7]. However, few vaccines are available to protect children, especially in the age group of 6 months to 2 years in selected regions, leaving unimmunized children vulnerable to COVID-19, along with a risk of further transmission to others. Furthermore, currently available vaccines require frequent booster doses to maintain protection and may cause adverse events such as headache, fever and mild to moderate reactions at the injection site [8,9]. As SARS-CoV-2 variants continue to emerge and cause infections with varying pathogenicity, there is a concern that currently available vaccines, which are mostly based on the spike protein of SARS-CoV-2, may not confer continued protection against COVID-19.

SARS-CoV-2 is a single-stranded positive-sense RNA virus belonging to the family *Coronaviridae*. The virus genome expresses 4 structural proteins (spike, membrane, envelope and nucleocapsid) and at least 16 non-structural proteins. The virus utilizes the receptor binding domain (RBD) of the spike protein to bind to the human Angiotensin Converting Enzyme-2 (ACE-2) receptor to infect human cells. Spike protein is implicated in causing severe infections and is the main target for neutralizing antibodies. Hence, the majority of COVID-19 vaccines, including vectored vaccines or novel-platform-based vaccines such as mRNA and DNA, utilize the SARS-CoV-2 spike protein and particularly the RBD as a basis for vaccine development. However, many T-cell and B-cell epitopes have been identified within the nucleoprotein and are spread over the entire proteome that play an important role in virus clearance and immunological memory. Currently available vaccines remain effective for a short duration and require frequent boosters. This could potentially lead to the resurgence of new variants [10,11]. Live attenuated vaccines (LAVs) offer several advantages over other vaccine platforms, including inactivated or subunit vaccines. LAVs contain a complete battery of viral antigens that induce both humoral and cellular immunity and provide high level of protection. Hence, LAVs are the most appropriate alternative to the existing vaccines to achieve a broader and more durable protective immune response [12,13].

Development of an adequately attenuated LAV strain that is not only safe but also immunogenic poses a major challenge. Furthermore, attenuation of traditional LAVs is generally based on a few specific point mutations (amino acid changes) introduced into the genome that may revert during replication. If the mutations are located in the region covering neutralizing epitopes, the vaccines may be rendered ineffective. To overcome these problems, a “Synthetic Attenuated Virus Engineering (SAVE)” technology platform was developed [14]. SAVE utilizes a computer-algorithm-directed “de-optimization” of the codon pairs of viral genes to introduce hundreds of silent mutations into the genome such that the modified codons are underrepresented in human cells. The resultant viral genome is de-optimized for protein translation in the human host cells. Using SAVE platform vaccines against polio, Zika and respiratory syncytial viruses have been found to be safe and effective in pre-clinical studies [14,15,16,17]. The SARS-CoV-2 live attenuated vaccine candidate (dCoV) used in the present study was designed using the SAVE platform and contains 283 silent mutations in the spike protein of the WA/1 strain and a 36-nucleotide (12-amino acid) deletion corresponding to the polybasic furin cleavage site between the S1 and S2 spike region for additional safety (Figure 1). We previously characterized the dCoV under the name COVI-VAC for attenuation markers. The safety of dCoV delivered through the intranasal (IN) route has been established in pre-clinical study in hamsters [18]. Further, the dCoV administered via IN route was found safe and immunogenic and demonstrated significant induction of T-cell response in the phase 1 clinical study conducted in healthy adults between 18–30 years (NCT04619628, Manuscript in preparation). The vaccine is being evaluated in a WHO sponsored phase 2/3 solidarity trial in multiple countries.

Given the high protective efficacy of the injectable live attenuated vaccines against respiratory infections such as measles, mumps and rubella, we tested the dCoV and the Measles-Rubella-dCoV (MR-dCoV) combination vaccine via the intramuscular (IM) route in hamsters for safety, immunogenicity and protective efficacy against the homologous (Wuhan) strain and heterologous (Delta) variant. The present schedule for the measles–rubella (MR) vaccine under the universal immunization program in India involves a first dose at 9–12 months of age and second at 16–24 months of age. Hence, the combination vaccine would circumvent the need for an additional injection in an already crowded pediatric vaccination schedule and would protect the children below the age of 2 years, an age group presently not covered by existing COVID-19 vaccines.

To study the immunogenicity and efficacy of the dCoV, we designed a challenge study in Syrian golden hamsters (*Mesocricetus auratus*). Hamster is a well-established model for the pathogenesis of COVID-19 and has a homologous ACE2 receptor to that of human [19,20]. SARS-CoV-2 infection in the hamsters peaked around 4–5 days. The infection was self-limiting and demonstrated mild to moderate disease, progressive weight loss, lung pathology associated with inflammation, interstitial pneumonia, hemorrhages, cellular infiltration and cytokine activation [19].

## 2. Materials and Methods

### 2.1. Production of Vaccine Strains and Challenge Viruses

To generate the dCoV vaccine, Vero cells (ATCC- CCL-81) were infected with the dCoV working seed virus, harvested in the supernatant and purified. Downstream processing involved clarification, nuclease treatment and tangential flow filtration to remove process impurities. The virus was stabilized, filtered through a 0.2 μm filter and stored below −60 °C.

To generate measles and rubella virus vaccines, the Edmonston Zagreb vaccine strain of the measles virus and the Wistar RA27/3 vaccine strain of the rubella virus were grown on MRC-5 cells; these components are part of the commercial measles–rubella (MR) vaccine produced by the Serum Institute of India Pvt Ltd (SIIPL). The combination vaccine was formulated using ≥ 3.0 Log_10_ CCID_50_ of measles and rubella vaccine components along with 5.0 Log_10_ PFU of the dCoV (in a log proportion of 3:3:5), stabilized with gelatin and sorbitol, lyophilized, and stored at 2–8 °C.

The MR vaccine used was from a commercial lot. The base medium containing gelatin and sorbitol stabilizer was used as a placebo. The wild-type SARS-CoV-2 isolates USA-WA1/2020 (Wuhan-like, 10^5^ TCID_50_/animal) and hCoV-19/USA/PHC658/2021 (Delta strain, lineage B.1.617.2, 10^4.3^ TCID_50_/animal) obtained from Biodefense and Emerging Infections Research Resources Repository, Virginia, USA was used for the challenge in hamsters. All virus strains used in the study were thoroughly characterized and sequenced using Sanger sequencing.

### 2.2. Determination of Vaccine Strains Titers via Plaque and CCID_50_ Assay

The dCoV vaccine strain was titrated using a standard plaque assay. Briefly, a Vero cell monolayer was prepared in 6-well plates one day prior to virus titration. A ten-fold serial dilution of the samples were prepared in Dulbecco’s Minimum Essential Medium (DMEM) containing 2% fetal bovine serum (FBS). The cell monolayer was washed once with DMEM and infected with serially diluted samples. After 1 h of incubation at 37 °C, 4.0 mL of overlay medium (1.25% carboxymethyl cellulose in DMEM containing 2% FBS) was added on the top of inoculums and further incubated at 37 °C undisturbed. After 3 days, the cells were washed with phosphate-buffered saline (PBS) and the plaques were visualized via negative staining with 0.3% crystal violet in 5% formalin. Excess stain was washed off using water. Distinctly countable plaques (between 20 to 120 plaques per well) were counted and the virus titer was calculated per the method described by Darling et al., 1998. Virus titers were expressed as Log_10_ plaque-forming units per 0.5 mL (Log_10_ PFU/0.5 mL) or per dose as applicable.

Measles and rubella viruses were titrated using the standard CCID_50_ assay. Measles virus was titrated using Vero cells plated on the day of test while rubella virus was titrated in RK-13 cells plated one day prior to test in 96-well plates. After infection with 10-fold serially diluted samples, the 96-well plates were incubated and observed microscopically on day 7 through day 10 post-infection for the cytopathic effect (CPE) indicating the presence of virus. The 50% end point was calculated using the Spearman and Karber method and the titers were represented as Log_10_ CCID_50_/dose.

### 2.3. Determination of Neutralization Titers by PRNT and MNT

The neutralizing antibodies titer against the dCoV was evaluated using the plaque reduction neutralization test (PRNT). Sera samples were pre-diluted to 1:5, followed by 4-fold dilutions in the diluent (DMEM containing 2% FBS). Each sera dilution was mixed with equal volume of 500 PFU/mL of the dCoV. In parallel, virus control was diluted 1:2 with the diluent. The virus–sera/diluent mixture was incubated at 37 ± 1 °C for 1 h for neutralization, and after incubation, 200 μL of the mixture was added onto a Vero cell monolayer in 24-well plates. After 1 h of incubation at 37 ± 1 °C, 1 mL of overlay media (1% carboxymethyl cellulose in diluent) was added in each well and the plates were incubated for three days at 37 ± 1 °C. The number of plaques was enumerated and 50% neutralization of the challenge virus (PRNT_50_) was calculated via probit analysis. Samples showing no virus neutralization were reported as titers < 10 and a value of 5 used for calculation of the geometric mean titer (GMT).

The neutralizing antibodies titer against MR was evaluated using the microneutralization test (MNT). Sera samples were diluted 1:10 followed by two-fold serial dilutions in Minimum Essential Medium (MEM). The sera dilutions were mixed with equal volume of 100 CCID_50_/mL of the measles virus or rubella virus and kept for neutralization at room temperature for 1.5 h. The measles virus–sera dilution mixtures were added onto Vero cell monolayers while rubella virus–sera dilution was added to RK-13 cells and incubated for up to 10 days at 36 ± 1 °C, after which the results were read via microscopic observations for CPE. The titers were calculated using the Spearman and Karber method and reported as 50% neutralization. Samples showing no virus neutralization were reported as titers < 10 and 5 for calculating GMT.

### 2.4. Evaluation of Biodistribution of Vaccine Strains in Hamsters

Virus replication and virus dissemination to different organs were studied in hamsters. Hamsters 4–5 weeks old were quarantined for 1 week prior to experimentation. The hamsters were divided into 4 treatment groups (*n* = 9/group): G1 (IN group), G2 (IM high-dose group), G3 (IM low-dose group) and G4 (placebo group). The animals weighing between 80–120 g were inoculated with a single dose of the vaccine via the IN route in a volume of 10 μL or IM route in a volume of 0.5 mL divided in two sites. In addition, animals were inoculated with normal saline (placebo) as the control group. Following IM injections, the sites were cleaned using 70% isopropyl alcohol. Hamsters tend to groom themselves by licking the site of inoculation. Therefore, to rule out the possibility of accidental infection via IN route, additional uninoculated animals (*n* = 6/group) were co-housed along with the IM groups (G2 and G3). Animal health and weight were taken throughout the observation period. Three inoculated animals/group (and two co-housed animals as applicable) per time point on day 3, day 6 and day 28 were sacrificed via overdose of isoflurane. Blood and vital organs including lungs, trachea, brain, heart, spleen and kidney were collected and weighed. Half of each organ was processed for viral load estimation while the other half was processed for histopathology. The organs were weighed and triturated with bio masher, and tissue suspensions in DMEM with 2% FBS were used for the detection of virus via standard plaque assay and real-time RT-PCR.

For real-time RT-PCR, viral RNA was extracted and detected using a very sensitive real-time RT-PCR based diagnostic platform (Xpert^®^ Xpress SARS-CoV-2) that detected the viral nucleocapsid (N2) and envelope (E) genes of SARS-CoV-2. The E gene provides good specificity while the N2 gene provides higher detection sensitivity. The assay detection limit was established using an internal reference virus of known titers. The cut-off Ct value of 36 for the E gene was equivalent to 0.01 PFU while the cut-off Ct value of 43 for the N2 gene was equivalent to 0.001 PFU. The assay failed to detect any virus beyond these cut-off values. All the animal studies were approved by the institutional animal ethical committee and institutional biosafety committee.

### 2.5. Challenge Studies in Syrian Golden Hamsters

#### 2.5.1. Immunization

Female hamsters of 8–10 weeks old (*n* = 8/group) weighing between 80–100 g were inoculated with 2 doses of vaccines (dCoV, MR or MR-dCoV) or placebo on day 0 and day 60 through IM route. Each dose of MR or MR-dCoV contained ≥3.0 Log_10_ CCID_50_/dose of the measles and rubella virus in line with the existing commercial MR vaccine. The amount of dCoV in the monovalent vaccine and the MR-dCoV vaccine was set at 5.0 Log_10_ PFU/dose (equivalent to the low dose in the biodistribution study) in the challenge study. The animals were monitored for temperature, body weight and food intake regularly during the immunization period. In life-bleeding (0.25 mL/animal) was performed via retro-orbital bleeding under isoflurane anesthesia to collect blood. The serum was separated from the collected blood by centrifuging at 4500 rpm for 10 min at 4 °C and stored below −20 °C until being tested for neutralizing antibody response.

#### 2.5.2. SARS-CoV-2 Challenge

The hamsters were transferred to a biosafety level 3 facility for acclimatization for 7 days prior to the SARS-CoV-2 challenge. On day 90, the animals were anesthetized by injecting 150 mg/kg ketamine and 10 mg/kg xylazine via the intraperitoneal route and challenged with wild-type SARS-CoV-2 isolate USA-WA1/2020 (Wuhan-like, 10^5^ TCID_50_/animal) and hCoV-19/USA/PHC658/2021 (Delta variant, lineage B.1.617.2, 10^4.3^ TCID_50_/animal) via the IN route. Post-challenge, all the animals were monitored daily for adverse clinical symptoms, body temperature, weight and feed intake.

Nasal swabs (from day 91 through day 94) were collected to assess virus shedding. For nasal swab collection, the swab was moistened in 1.0 mL of serum-free media and used to rub the outside of the hamster nose. The swab was placed into a vial containing 200 μL of medium, mixed and then pressed against the wall of the tube to drain the medium from the swab. The nasal swab collections were stored at −80 °C for subsequent virological analysis.

Blood was collected on day 94 (4 days post-challenge) via the retro-orbital route under isoflurane anesthesia before euthanization. After euthanization, the lungs were aseptically removed from all animals to determine the viral load. Gross pathological examination was performed and the lungs were weighed. Representative lungs from each group were photographed to document observations. Left lobes were perfused with 10% neutral buffered formalin for histopathological examination while right lobes were homogenized using Pro 200 homogenizer (Pro Scientific Inc., Monroe, CT, USA) in 1.0 mL sterile PBS.

The viral RNA was quantified via qRT-PCR assay. RNA was extracted from the nasal swabs and lung homogenates using a QIAamp Viral RNA Mini Kit (Qiagen, Rrevised. Germantown, MD, USA). qRT-PCR assay was performed using a US-2019-nCoV assay kit (Sigma, Cat No. CDA00011), 2019-nCoV_N1 forward primer (5′-GACCCCAAAATCAGCGAAAT-3′), and 2019-nCoV_N1 reverse primer (5′-TCTGGTTACTGCCAGTTGAATCTG-3′) and probe (5′-FAM-ACCCCGCATTACGTTTGGTGGACC-BHQ1-3′). The positive control RNA from heat-inactivated SARS-related coronavirus 2, isolate USA-WA1/2020 (BEI resources, Cat No.: NR-52347) was used to generate the standard curve. The SARS-CoV-2 gene copies per lung were calculated to determine the log reduction of viral RNA in the immunized group compared to the infection control group.

For estimation of the live virus in lungs via TCID_50_ assay, approximately 30,000 Vero E6 cells were plated in a volume of 200 μL/well in DMEM containing 10% FBS into 96-well plates and incubated overnight (12–18 h) at 37 °C to achieve a cell monolayer. The lung homogenate samples were serially diluted (10-fold) in DMEM and 50 μL of each dilution was plated in quadruplicate wells of a 96-well plate containing a cell monolayer. Plates were incubated at 37 °C with 5% CO_2_ for 1 h with shaking at every 15 min. Next, the wells were supplemented with 150 μL of DMEM and further incubated to detect CPE on day 5 for the Wuhan strain and day 7 for the Delta variant. To visualize CPE, DMEM was gently removed from the plates with a pipette and washed twice with PBS. Cells were fixed with 200 μL of 4% formaldehyde per well for 30 min at room temperature and stained with 0.05% (*w*/*v*) crystal violet for 30 min. Upon completion of staining, crystal violet was removed with a pipette and washed twice with distilled water or until excess crystal violet was removed and CPE was clearly visualized. The plates were scored for positive wells showing CPE and the virus titer in each sample was calculated using the Reed and Muench formula. The viral load per g of lung was calculated.

### 2.6. Statistical Analysis of the Data

Where applicable, results were plotted using GraphPad prism v 7.05 and GMT and standard deviation was calculated. For comparison between two or more groups, student’s *t* test or ANOVA at a level of significance of 0.05% was calculated. Where applicable, animal weights were compared within each group using one- or two-way analysis of variance (ANOVA), followed by Tukey’s test to compare multiple groups or Dunnett’s multiple comparisons test to compare each group with the control as applicable.

## 3. Results

### 3.1. Safety Evaluation and Bio-Distribution of dCoV in Hamsters

The safety and efficacy of LAVs is largely dependent on the route of administration, virus replication and virus dissemination to different organs.

To elucidate the bio-distribution of the dCoV, the vaccine was inoculated in three groups of hamsters (*n* = 9/group): IN route at 6.0 Log_10_ PFU/animal (G1), IM route at 6.0 Log_10_ PFU/animal (G2, high dose) and 5.0 Log_10_ PFU/animal (G3, low dose). An additional six un-inoculated hamsters were co-housed for both G2 and G3. Control group hamsters were inoculated with normal saline through the IM route (G4, placebo, *n* = 9) (Figure 2A). None of these animals demonstrated disease symptoms or loss in body weight and remained healthy for the observation period of 28 days (Figure 2B). Three animals from G1–G4 and two co-housed animals from both G2 and G3 were sacrificed on each time point of day 3, day 6 and day 28. Blood from all the animals was collected prior to sacrifice. Trachea, lungs, brain, kidney, spleen and heart were collected from each hamster. Sera samples and triturated tissue samples (approx. 200 mg of each organ suspended in 500 μL of DMEM containing 2% FBS) were evaluated for viral load via detection of two viral genes (E gene, N2 gene with high sensitivity of detection) of the dCoV using qRT-PCR and infective virus particles via plaque assay.

In the IN group (G1), dCoV RNA was detected on day 3 in the sera (N2 gene) and in most of the organs (E and N2 gene) of all animals, with lower Ct in the trachea and lung samples. Reduction in the viral RNA could be seen by day 6 and nearly complete clearance by day 28 with trace amount of the viral RNA (only N2 gene) detected in the trachea of two animals and the lung of one animal (Figure 2C). Further, sera and tissue samples were subjected to plaque assay to detect live virus. No live virus was detected in the sera from any animal. Live vaccine virus was detected only in trachea of all hamsters in the G1 group on day 3 post-administration (viral load = 4.41, 2.52 and 3.97 Log_10_ PFU/g) and subsequently, on day 6 (*n* = 3) and day 9 (*n* = 3), no live virus was detected (Figure 2D). After further virus amplification by blind passage of negative samples in cell culture, virus could only be detected in lungs from the G1 group, indicating presence of a low level of infective virus on day 3. All other organs were free of the live virus, despite a low level of viral RNA being observed in the organs.

In the IM high-dose group (G2), only the N2 gene was detected in sera samples on day 3 in all animals. In contrast, only one animal demonstrated the presence of viral RNA (E and N2 gene) in most of the organs except the brain on day 3 and day 6. Viral RNA was completely cleared by day 28 and no live virus was detected in sera or organs via plaque assay.

In the IM low-dose group (G3), no viral RNA was detected in sera samples. Viral RNA (N2 gene) was detected only in the trachea of one animal on day 3. No viral RNA or live virus particles could be detected in the sera or organs on day 6 and day 28, indicating lack of replicating virus in these samples (Figure 2C). Placebo (G4) and co-housed animals (in G2 and G3 groups) were negative for the viral RNA and live virus in sera and all organs.

Overall, the data indicate that active virus replication following vaccine administration via the IN route was restricted to the trachea, whereas no active virus replication could be detected when the vaccine was administered via the IM route. The dCoV was found safe via both routes and did not show any safety concerns when tested at the high-dose level (6.0 Log_10_ PFU), which was around 10 times higher of a dose than the intended human dose. Histopathological examination of organs collected on day 6 and day 28 post-inoculation during the safety study were similar to that of the placebo group. Sparse mononuclear cell infiltrate in the trachea was observed in all animals. Similarly, polymorphs were observed in the lungs, heart and kidney in all hamsters and could not be attributed to any specific group (Appendix A).

### 3.2. SARS-CoV-2 Challenge Study Demonstrated Protective Efficacy of Vaccines

#### 3.2.1. Safety and Immunogenicity of the dCoV

The virus challenge studies were conducted in two sets—A. challenge with the Wuhan-like strain and B. challenge with the Delta variant (Figure 3A). Hamsters (*n* = 8/group) were administered two doses of the vaccine/controls 60 days apart and then challenged on day 90, and animals were monitored until day 94. Temperature monitoring throughout the experimental period did not show rise in temperature in the vaccinated animals (Appendix A), and all animals demonstrated an increase in body weight during the immunization/pre-challenge phase (Figure 3B,C) indicating that the vaccine is well tolerated through both IN and IM routes.

Pooled sera samples from each group of hamsters (*n* = 8) collected on day 0, day 30, day 60 and day 90 were evaluated for neutralizing antibody titers against the Wuhan-like strain and the Delta variant using PRNT assay. Day 30 and day 60 post-one dose, animals immunized with the MR-dCoV or the dCoV exhibited seroconversion against both the Wuhan-like strain and the Delta variant. PRNT titers on day 60 remained similar to titers obtained on day 30 (after one dose) and were boosted to around twofold on day 90. The dCoV administered via the IN route demonstrated significantly elevated level of antibody response while titers in the placebo group remained at baseline (Figure 3D). In animals inoculated via the IM route, slightly higher titers were observed in the MR-dCoV group compared to the dCoV group. Considering the small number of animals, it cannot be surmised whether a synergistic effect contributed to observed titers.

Further, evaluation of neutralizing antibody titers via MNT assay for measles and rubella components in pooled sera samples collected on day 90 from animals immunized with MR-dCoV, MR or placebo via the IM route exhibited similar titers in both vaccine groups (Figure 3F). This suggests no interference among viral components of the MR-dCoV combination vaccine in inducing antibody response against individual virus components.

Following IN challenge with the wild-type virus, transient declines in body weights were observed in all groups, which was consistent with the historical lab data (Figure 3B,C). This could be attributed to transient decline in feed consumption observed during challenge phase in all groups (Appendix A).

#### 3.2.2. Protection against Wuhan Challenge

After challenge with wild-type SARS-CoV-2 Wuhan-like strain, peak viral RNA in nasal swabs of the infection control group (placebo) was between 7.08–7.77 Log_10_ copies/mL from 1 day post-infection (dpi) to 4 dpi. The animals immunized with the dCoV via IN route exhibited significant reduction in viral RNA on all four days, ranging from 2.96–3.75 Log_10_ copies/mL. Animals immunized via the IM route with the dCoV or the MR-dCoV vaccines demonstrated reduced RNA copies (1.0 Log_10_ copies/mL), which were statistically significant only on day 3 and day 4 post-challenge. In contrast, MR vaccinated hamsters demonstrated the presence of viral RNA in nasal swabs on all four days post-challenge and the copy number ranged from 7.08–7.77 Log_10_ copies/mL, similar to the placebo group (Appendix A).

The lungs of animals immunized with the MR vaccine or the placebo (infection controls) demonstrated an average RNA copy number of 8.53 Log_10_ copies/lung and an average infectious virus titer of 6.78 Log_10_ TCID_50_/lung. Significant reduction in the viral RNA of 3.14 and 2.65 Log_10_ copies/lung were observed in the animals immunized with the MR-dCoV and the dCoV, respectively (Figure 4B). Since RNA detection using qRT-PCR is known to detect live as well as dead/defective virus particles, these samples were also subjected to detection of infectious virus in lung suspensions via TCID_50_ assay. Complete protection from the replicating Wuhan-like challenge strain was observed in the lungs of hamsters immunized with the MR-dCoV and the dCoV (Figure 4C).

#### 3.2.3. Protection against Delta Challenge

The animals immunized with the MR-dCoV/placebo through the IM route and the dCoV through the IN route were challenged with the Delta variant. The peak viral RNA of the Delta variant ranged from 6.18–8.21 Log_10_ copies/mL in nasal swabs from 1 dpi to 4 dpi (Appendix A). Though animals immunized with the MR-dCoV via IM route did not show significant reduction in viral RNA copies in nasal swabs, significant reduction was observed in the animals immunized with dCoV via the IN route.

The average RNA copy number in the lungs of the infection control with placebo group was 9.15 Log_10_ copies/lung. The log reduction of viral RNA was found to be 2.11 Log_10_ copies/lung and 5.40 Log_10_ copies/lung in the animals immunized with the MR-dCoV (IM) and dCoV (IN) vaccines, respectively (Figure 4E). Infectious virus particles estimated via TCID_50_ assay demonstrated an average viral load of 5.13 Log_10_ TCID_50_/lung in the placebo. However, no infectious virus particles could be detected in the lungs of animals immunized with the dCoV (IN) and the MR-dCoV (IM), indicating complete protection from lung infection in vaccinated groups (Figure 4F).

Overall, immunization with the dCoV as a monovalent vaccine or in the MR-dCoV combination vaccine was able to protect hamsters from lung damage and virus spread following challenge with the homologous strain (Wuhan) and the heterologous strain (Delta).

#### 3.2.4. Histopathology Evaluation

Animals challenged with Wuhan strain: The gross pathological examination of lungs found that animals immunized with the MR vaccine alone or placebo (infection control) challenged with the Wuhan-like strain had moderate inflammation and edema along with multifocal hemorrhages. In contrast, lungs collected from challenged animals immunized with the MR-dCoV through the IM route and dCoV through both IM and IN routes appeared healthy (Figure 4A). The histopathology of lungs from infection control animals demonstrated mild to moderate multifocal broncho-interstitial pneumonia with alveolar infiltration of inflammatory cells such as lymphocytes and macrophages, occasional neutrophils and heterophils, and focal hemorrhages. Furthermore, moderate and multifocal bronchial epithelial degeneration, hyperplasia endotheliaitis with perivascular edema, cuffing with mononuclear cells, and proliferation of type-II pneumocytes was also observed. However, no lesions were observed in 50% of animals immunized with the MR-dCoV (IM route) and the dCoV (IM and IN routes), while the remaining 50% exhibited signs of minimal broncho-interstitial pneumonia and inflammatory changes. In contrast, mild to moderate inflammatory changes were observed in challenged animals previously immunized with the MR vaccine (Figure 5).

Animals challenged with Delta variant: The gross pathology of lungs from animals (infection control group) challenged with the Delta variant exhibited mild generalized edema with areas of focal congestion at 4 dpi (Figure 4D). However, the MR-dCoV (IM) and the dCoV (IN) group exhibited apparently normal lungs (Figure 4D). The histopathology study found mild multifocal broncho-interstitial pneumonia and bronchial epithelial changes such as degeneration, hyperplasia, alveolar and bronchial infiltration of inflammatory cells along with alveolar edema and hyperplasia of type II pneumocytes in the infection control group. However, the group immunized with the dCoV (IN) exhibited no lesions, while the MR-dCoV (IM) group exhibited no lesions to minimal inflammatory changes (Figure 5).

## 4. Discussion

Our primary objective is to develop a vaccine to immunize and protect children below 2 years of age. Considering a large-scale vaccine uptake globally, it is expected that the majority of women of child-bearing age have a considerable amount of anti-SARS-CoV-2 antibodies. These antibodies are transferred to infants as passive immunity like other vaccines [21,22,23]. These passively transferred antibodies will wane within the first 6 months of life of the infant. We, therefore, propose to immunize infants beyond 6 months of age and children using LAVs.

COVID-19 was rare among children due to closure of schools, home quarantines and less testing in earlier days of the pandemic [4]. During initial waves of the COVID-19 pandemic, children were not covered in vaccination campaigns. Moreover, the virus is rapidly evolving to new SARS-CoV-2 variants with uncertainty regarding morbidity and transmissibility. Therefore, children must be covered in vaccination efforts with a sustainable vaccine that would provide a broader immune response to counter the strain variation. Currently, licensed vaccines rely on the immune response against only the spike protein on SARS-CoV-2, which results in reduced efficacy and waning of antibody levels over time, requiring repeated booster doses [7,24,25,26]. In contrast, immunity generated by the whole virus would be evidenced to be more robust and long-lasting. In support of this argument, a recent study from Singapore reported a long-lasting antibody response with at least 13 months of longevity following mild COVID-19 disease [27]. This indicates that a LAV that induces the immune system like a wild-type infection would also provide long-lasting immunity to COVID-19.

Historically, LAVs have been evidenced to be effective compared with inactivated or subunit vaccines because LAVs elicit a broader immune response. The LAV used in this study (dCoV) is based on codon pair de-optimization. The vaccine strain was attenuated synthetically by incorporating two features to ensure a high level of safety without compromising the efficacy: 1. introduction of computationally identified multiple silent mutations in the spike gene to achieve optimal attenuation, and 2. deletion of 36 nucleotides that encode 12 amino acids in the furin cleavage site between the S1 and S2 domains of the spike protein. Therefore, sub-optimal translation resulted in limited virus replication [14,18]. Using this codon pair de-optimized strain, we developed two dCoV vaccine formulations, one suitable for administration via the IN route and other via the IM route. The third formulation, developed as a Measles-Rubella-dCoV combination vaccine (MR-dCoV), is intended for administration via the IM route as an alternative to the existing MR vaccine. The safety and protective efficacy of the injectable MR vaccine against measles and rubella that spread through the aerosol route has been well established in infants, children, adolescents and adults [28,29]. The MR-dCoV combination vaccine would not require an additional prick or an additional visit to the clinic and render effective protection against emerging variants of the SARS-CoV-2 in the younger age group within the routine childhood vaccination schedule. Therefore, vaccination with the MR-dCoV combination vaccine would lead to vaccine compliance without introducing an additional vaccine in the already crowded vaccination schedule for children. In India, MR vaccination is given under the Universal Immunisation Programme at 9–12 months of age and the second dose at 16–24 months of age. Hence, a longer interval of 60 days between two doses followed by a challenge on day 90 was explored in our hamster study. A mumps vectored vaccine has been demonstrated to work well in animal models [30]. However, use of a vaccine strain with established safety and efficacy would be relatively quicker to deploy for routine immunization compared to new vectored vaccines.

Transient virus replication is a hallmark of LAVs and is required for efficient immune activation. We conducted bio-distribution and challenge studies in hamsters since these animals are a reliable model to study pathogenesis [31,32]. In hamsters, the peak infectious virus load of wild-type coronavirus in lungs is reported between day 2 to day 7 at around 7.0 Log_10_ TCID_50_/mL (or 8–10 Log_10_ TCID_50_/lung) [32,33]. Our bio-distribution data in the hamsters show that the dCoV administration via the IN route was infectious since the dCoV could be detected locally in the trachea on day 3, with low levels of viral RNA. Transient virus replication was restricted in the lower respiratory tract, as a very low level of the live virus was detectable in the lungs. In contrast, control group hamsters challenged with 6.0 Log_10_ PFU wild-type Wuhan-like demonstrated high levels of live virus in lungs on day 3 post-challenge (6.78 Log_10_ TCID_50_/mL), attesting to the attenuated phenotype of the vaccine strain (see Figure 4C for wild-type comparison). We did not detect infective dCoV in subsequent sampling performed on day 6 and day 28, though a low level of RNA was detected in a few samples on subsequent days, indicating limited virus replication in the lungs and trachea that did not spread to other organs. Further, animals administered with the vaccine via the IM route did not show active virus replication (live virus) in any of the vital organs. This observation was supported by the low level of RNA detected in animals administered with the high dose (10 times the intended human dose) [34]. Similarly, no live virus was detected in the group receiving the low dose.

Our study design also included co-housed animals alongside the IM groups as a precautionary control. Since hamsters tend to sniff and groom the inoculated sites, hence, when the animals were given the dCoV vaccine via the IM routes, there was a theoretical possibility of cross-infection via the IN route. None of the co-housed animals exhibited presence of virus in any of organs, indicating no cross-infection via the IN route. We did not find any effect on the animal weights, health or histopathology. No infectious virus could be detected in any of the organs when the vaccine was administered via the IM route, thus making it a safer alternative for adults and children. The dCoV was previously tested as a booster vaccine both via the IM and IN route (manuscript in preparation). In the booster study, two placebo groups were included as controls for both IM and IN routes. No safety concern in the vaccinated or placebo groups was observed and the neutralizing antibody titers in the placebo groups remained at a baseline.

In our challenge studies, we immunized hamsters with the MR-dCoV and the dCoV along with MR and placebo as controls. In this study, protective efficacy was established at a 10 times lower dose level of the dCoV component than the level at which safety was established in bio-distribution studies. In view of the available data on the IN route [18] a direct comparison of the IN and IM routes was incorporated in the study design. We observed that a single dose of the dCoV vaccine administered through both routes resulted in the production of neutralizing antibodies, which were further boosted after the second dose. We did not find any vaccine-induced adverse events throughout 90 days of observation period. We studied the virus dissemination by quantifying viral RNA in nasal swabs collected from 1 dpi to 4 dpi with the wild-type viruses (Wuhan-like strain or Delta variant). We observed significant reduction in the viral RNA levels in animals immunized with either the MR-dCoV or monovalent dCoV vaccine on day 3 and day 4 post challenge with the Wuhan-like strain, though the results were relatively less prominent with the Delta variant. As expected, significant reduction in the viral RNA was observed throughout 1 dpi to 4 dpi in animals immunized through IN route. Further, significant reduction in the viral RNA (Wuhan-like and Delta) was found in the lungs of animals immunized with the MR-dCoV or dCoV vaccine when compared with the infection controls (placebo or MR). In support, no infectious virus could be detected in the immunized animals. Thus, our study results strongly indicate that the vaccine could effectively inhibit virus dissemination and offer complete protection against both the Wuhan-like and Delta strains.

For challenge, we used the SARS-CoV-2 Wuhan-like strain and Delta variant since these variants are often associated with serious disease and enhanced transmissibility. A recently circulating SARS-CoV-2 Omicron variant has been reported to exhibit reduced replication and fusion activity, causing mild symptoms in humans and in the animal model [35]. Nevertheless, effectiveness against the Omicron strain and newly emerging variants needs to be studied, considering the significant antigenic changes compared to the Wuhan strain. The present study shows that the candidate vaccine dCoV attenuated by codon pair de-optimization technology administered either via IN or IM routes was safe and effective against virulent SARS-CoV-2 challenge in hamsters. The protective efficacy from the MR-dCoV vaccine is evidenced by immunized hamsters appearing healthy with no pathological changes, while placebo controls challenged with the Wuhan-like strain or delta variant exhibited mild to moderate inflammation and edema along with multifocal hemorrhages.

In conclusion, the dCoV vaccine induced a sufficient level of neutralizing antibodies against the SARS-CoV-2 variants Wuhan-like SARS-CoV-2 isolate (USA-WA1/2020) and Delta variant (B.1.617.2). Hamsters immunized via both IN and IM routes with the dCoV vaccine exhibited complete protection against both variants of SARS-CoV-2 in the virus challenge study. In addition, a combination of dCoV with the presently available MR vaccine through the IM route also protected hamsters after exposure to a challenge virus. There was no interference of the additional dCoV to the existing measles and rubella vaccine. The injectable (IM) vaccine formulations are currently being evaluated in pre-clinical studies while the safety of the dCoV vaccine via IN route was established in a Phase 1 clinical trial in healthy adults between 18–30 years of age. Being a live, attenuated vaccine, this is an ideal candidate for a prime-boost strategy or simultaneous immunization wherein a vaccine can be primed through one route and boosted via an alternate route, providing protection against emerging SARS-CoV-2 variants.

## Figures and Tables

**Figure 1 vaccines-11-00255-f001:**
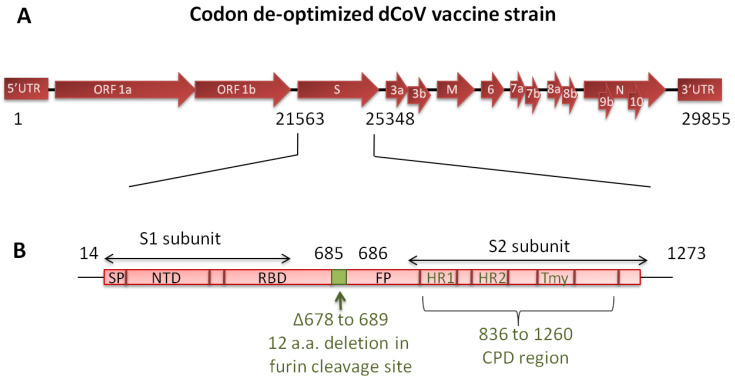
Schematic representation of the live attenuated SARS-CoV-2 vaccine strain (dCoV). (**A**) Genome consists of 29,855 nucleotides. Spike gene (21,563–25,348 nucleotides) contains 36 nucleotides deletion corresponding to 12 amino acids (Δ678 to 689) and 283 nucleotides mutation to achieve codon pair de-optimization. (**B**) Spike protein region and the relative positions of deletion of 12 amino acids at furin cleavage site and codon pair de-optimized region (CPD) after translation.

**Figure 2 vaccines-11-00255-f002:**
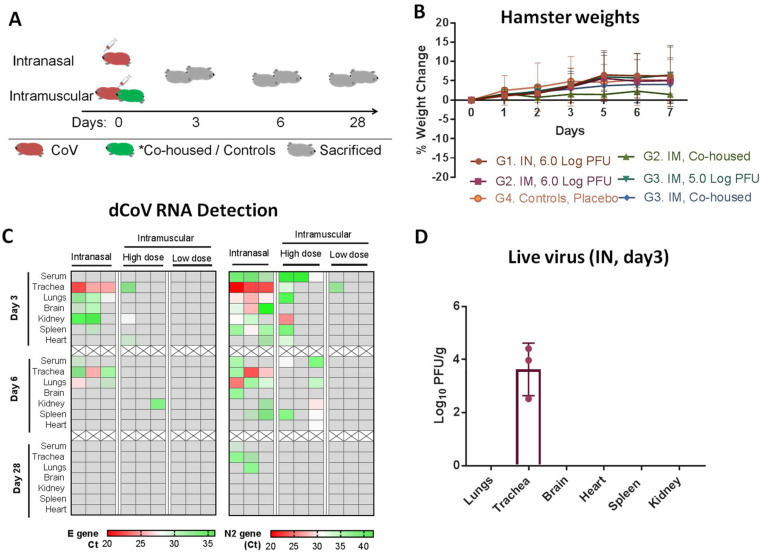
Safety and Biodistribution of CoV in Hamsters (**A**) Schematic representation of the experimental design. Hamsters (*n* = 9) per group, 4 groups were administered dCoV either via IN route (G1) or IM (G2—6.0 log_10_ PFU/animal (high dose); G3—5.0 Log_10_ PFU/animal (low dose)) (red color). Control animals (green) consist of co-housed animals (*), 2 hamsters per cage in IM groups G2 and G3) or negative controls (G4) that received placebo. Three animals per group were sacrificed on day 3, day 6 and day 28 post administration to study viral load. (**B**) Animals’ weights (gram) monitored during experimental period. (**C**) Heat map from each group representing viral RNA (E and N2 gene) detected using real time PCR. Grey cells represent no viral RNA detected (cut-off Ct value of E gene having high specificity is equivalent to detection limit of 0.01 PFU and N2 gene with high sensitivity had a detection limit equivalent 0.001 PFU). (**D**) Presence of live replicating virus in trachea detected on day 3 post IN administration of dCoV represented as a dot per animal, bar with Mean and SD (*n* = 3) is shown.

**Figure 3 vaccines-11-00255-f003:**
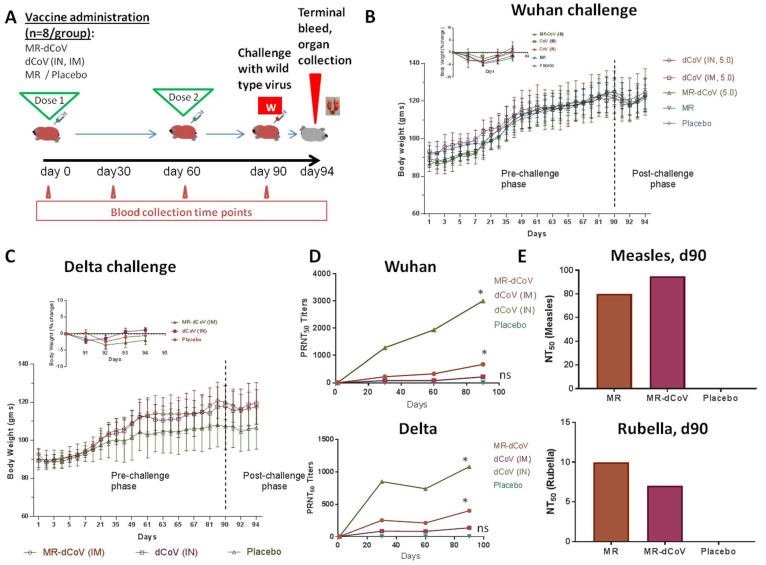
Challenge with Wild-type SARS-CoV-2 Variants: (**A**) Schematic representation of the experimental design of challenge studies. (**B**) Weights of hamsters in grams throughout the observation period and inset showing % weight change after challenge with the Wuhan strain from day 90 to day 94 (1–4 days post challenge). Line graph with Mean and SD is shown for each group, dCoV administered via IN routecircle) or via IM route (square), MR-dCoV (triangle), MR (inverted triangle) and Placebo (diamond). (**C**) Weights of hamsters in the challenge study of the Delta strain and inset showing % weight change after challenge with the Delta strain from day 90 to day 94 (1–4 days post challenge). Line graph with Mean and SD is shown for each group, MR-dCoV (triangle), dCoV (circle) and Placebo (inverted triangle). (**D**) Neutralizing antibody titers in hamster sera samples collected from day 0 to day 90 in the pre-challenge phase and tested against the Wuhan strain (top) or the Delta strain (bottom) estimated via PRNT assay. Rise in antibody titers was observed in the hamsters vaccinated with MR-CoV and dCoV (IN) with respect to placebo. One-tailed, paired Student’s *t* test, * ≤0.05, ns = no significant difference observed with respect to placebo controls. (**E**) Neutralizing antibody titers against measles virus and rubella virus in hamster sera on day 90 tested using MNT assay.

**Figure 4 vaccines-11-00255-f004:**
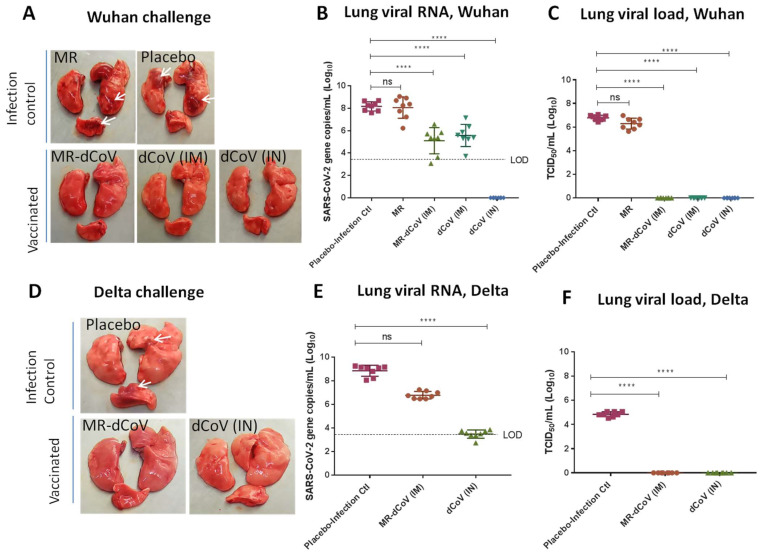
Protection in hamsters post-challenge with wild-type SARS-CoV-2 variants: (**A**–**C**) Challenge with Wuhan strain, (**A**) gross pathology of lungs from vaccinated (MR-dCoV or control dCoV) and unvaccinated (placebo). Multifocal hemorrhages (white arrows) are evident in the lungs from unvaccinated control groups. (**B**) Viral RNA detected via qRT-PCR. (**C**) Live virus detected via TCID_50_ assay. (**B**,**C**) Placebo-infection control (square), MR (circle), MR-dCoV (triangle), and dCoV administered via IM route (inverted triangle) and via IN route (diamond). (**D**–**F**) Challenge with Wuhan strain, (**D**) multifocal hemorrhages (white arrows) are evident in the lungs from unvaccinated control groups. (**E**) Viral load detected in the lungs on day 94. (**F**) Live virus detected in lungs via TCID_50_ assay. (**E**,**F**) Placebo-infection control (square), MR-dCoV (circle) and dCoV (triangle). Error bars = mean ± SD, *n* = 8, one-way ANOVA—Dunnett’s multiple comparison test, **** = *p* ≤ 0.0001, ns = no significant difference observed with respect to infection control groups.

**Figure 5 vaccines-11-00255-f005:**
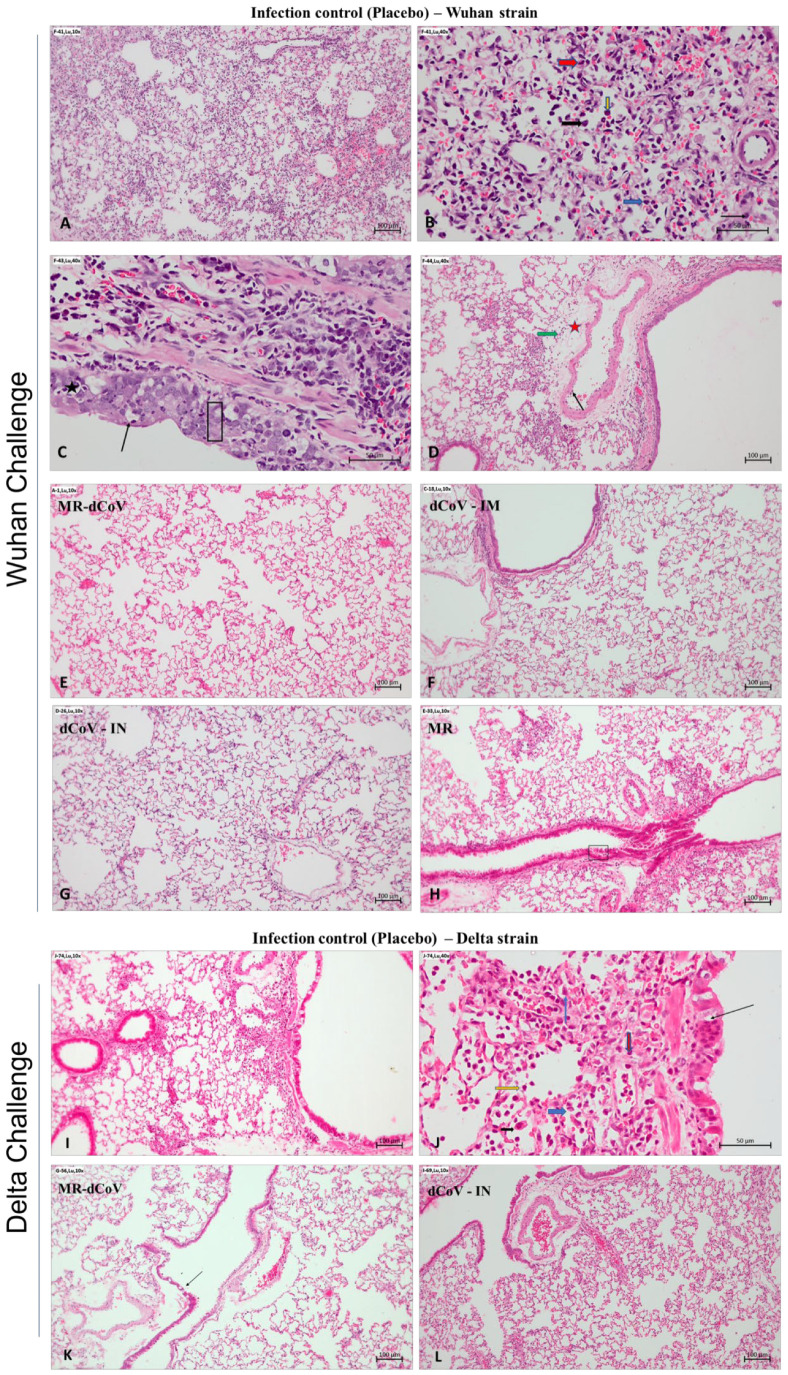
Histopathological observations of the lung challenged with SARS-CoV-2: (**A**–**H**) Challenge with SARS-CoV-2 Wuhan strain—(**A**–**D**) infection control (placebo), (**A**) moderate broncho-interstitial pneumonia with hemorrhages. (**B**) Moderate and multifocal alveolar damage and infiltration of lymphocytes (yellow arrow), macrophages (black arrow), occasional neutrophils (blue arrow) and heterophils (red arrow). (**C**) Bronchial epithelial degeneration (thin arrow) and necrosis (black star) with inflammatory cells, and hyperplasia of bronchial epithelial cells (rectangular area). (**D**) Perivascular edema (red star) and cuffing with mononuclear cells (green arrow). Endotheliaitis (thin black arrow). (**E**–**H**) Vaccinated group animal challenged with the Wuhan strain virus: (**E**) MR-dCoV immunized via IM route. Lungs appear normal. (**F**) dCoV immunized via IM route. Lungs are within normal limits, with bronchi showing epithelial hyperplasia. (**G**) dCoV immunized via IN route. Lungs appear normal (**H**) MR immunized via IM route. Lungs show mild focal inflammatory changes with hyperplasia of bronchial epithelial cells (black rectangle). (**I**–**L**) Challenge with the SARS-CoV-2 Delta variant: (**I**–**J**) Infection control (placebo), (**I**) broncho-interstitial pneumonia, minimal and multifocal. (**J**) Bronchial epithelial degeneration, minimal (thin black arrow); alveolar infiltration of inflammatory cells: lymphocytes (yellow arrow), macrophages (black arrow), occasional neutrophils (blue arrow) and heterophils (red arrow), as well as proliferation of type II pneumocytes (thin blue arrow), minimal. (**K**) MR-dCoV immunized via IM route. In lung, bronchial epithelial degeneration, minimal (thin black arrow). (**L**) dCoV immunized via IN route. Lungs appear normal.

## Data Availability

Data is contained within the article and Appendix A.

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
