# Peer review of "A Live Attenuated COVID-19 Candidate Vaccine for Children: Protection against SARS-CoV-2 Challenge in Hamsters"

_vaccines, 2023, doi:10.3390/vaccines11020255_

Round 1
Reviewer 1 Report
The manuscript aims to investigate the efficacy of a live attenuated vaccine candidate against SARS-CoV-2. The virus candidate was developed based on the concept of codon-deopmitization (SAVE platform) and tested for its safety and ability to elicit protective immune responses against SARS-CoV-2 in a hamster model. The authors demonstrated that the vaccine candidate was safe in animals and immunized animals could be protected from infection with wild-type virus. Overall, this study is well designed and the conclusion could be supported by the results presented in this paper. However, there are some concerns that should be addressed by the authors.
COMMENTS
1. Please clearly indicate whether the candidate vaccine tested in this paper is identical to the vaccine previously characterized by Wang et al (reference 18). If not, please indicate the difference and cite the characteristics of the virus, e.g., plaque size or growth kinetics.
2. Since the vaccine candidate is intended for use in children, how old is the hamster used in this study? Could the authors justify this point and address it in the manuscript.
3. The result shown in Fig. 3D indicates that hamsters immunized with the candidate vaccine gained more weight than the placebo group. What could be the explanation for this observation?
4. It is not clear from the manuscript how much weight the hamsters lost after the Wuhan or Delta Challenge. From the graph shown in Fig. 3B and 3C, it appears that the difference in weight loss is not significant. I would suggest the authors to provide more data, especially the percentage weight loss on each day after viral challenge.
5. Please explain why pooled serum was used to determine PRNT in Fig.3D. I would suggest that you provide the data with the statistical analysis performed.
6. The data showing no detectable infectious viruses in the lungs of challenged animals (in vaccinated groups) may be somewhat problematic. Based on the significant weight loss after viral challenge in all groups indicating clinical signs in the animals. The fact that no virus was found in the lungs after provocation seems to contradict this observation. I find it difficult to find these data convincing.
7. Some figures need to be corrected. The legend is cropped, missing important information.
Author Response
Response to Reviewer 1 Comments
Point 1. Please clearly indicate whether the candidate vaccine tested in this paper is identical to the vaccine previously characterized by Wang et al (reference 18). If not, please indicate the difference and cite the characteristics of the virus, e.g., plaque size or growth kinetics.
Response 1: We sincerely acknowledge the comment made by the reviewer. As pointed out, strain information has been incorporated in the introduction section to ensure clarity about the candidate vaccine. The candidate vaccine strain used in this paper was previously characterized for attenuation under the name COVI-VAC (Wang et al., 2021). In this manuscript, two vaccine formulations were used (dCoV vaccine and a combination vaccine MR-dCoV). In both formulations, identical vaccine candidate COVI-VAC was used as previously described by Wang et al., 2021.
Point 2: Since the vaccine candidate is intended for use in children, how old is the hamster used in this study? Could the authors justify this point and address it in the manuscript.
Response 2: We thank reviewer for the suggestion. For evaluation of dCoV biodistribution, 4-5 weeks old hamsters quarantined for 1 week. At the time of experimentation the animals were between 5-6 weeks of age and weighted between 80-120 g. Similarly, for challenge studies, 8-10 weeks old hamsters’ weighing in the range of 80-120 g was used. Material and method section has been updated to include age of hamsters.
Hamsters are very sensitive to SARS-CoV-2 infections. Upon infection, they quickly develop severe inflammation of lungs and pneumonia which is coupled with weight loss. To accurately detect weight loss animals in the above-mentioned weight range is required. The weight range has been used in by other investigators and has been well established (Sia S.F. et al, 2020; Yang SJ et al, 2021)
Hamster challenge studies were conducted only as a proof-of-concept for attenuated vaccine strains. The vaccine candidate shall be evaluated for safety in Phase 1/2 clinical trials in adults followed by children in an age de-escalation manner.
Point 3: The result shown in Fig. 3D indicates that hamsters immunized with the candidate vaccine gained more weight than the placebo group. What could be the explanation for this observation?
Response 3: We assume that the reviewer is referring to the figure 3C wherein the data for weight changes in hamsters has been presented. In our challenge study (Figure 3C), average weight change in hamsters groups immunized with the candidate vaccine was higher compared to placebo group. However, percent weight change in both groups was within 10% variation and was not statistically significant. Animals studies in hamsters are prone to variations in the observed range as these animals are very sensitive to weight changes. Typically hamsters loose 10-15% of body weight upon wild-type infection with SARS-CoV-2. More than 10% weigh change is considered significant and has been reported by other investigators (Yang SJ et al, 2021). In this study, we did not observe loss of weight in the vaccinated animals which clearly reiterate attenuation of candidate vaccine strain in the pre-challenge phase and protective efficacy of candidate vaccine following challenge with the wild type SARS-CoV-2.
Point 4: It is not clear from the manuscript how much weight the hamsters lost after the Wuhan or Delta Challenge. From the graph shown in Fig. 3B and 3C, it appears that the difference in weight loss is not significant. I would suggest the authors to provide more data, especially the percentage weight loss on each day after viral challenge.
Response 4: We thank reviewer for the constructive suggestion. The weight changes post-challenge was less than 5% after challenge with wild type either Wuhan or delta strain. Figure 3C has been updated to incorporate weight change post-challenge with delta strain. The legends have been updated accordingly.
Point 5: Please explain why pooled serum was used to determine PRNT in Fig.3D. I would suggest that you provide the data with the statistical analysis performed.
Response 5: We thank reviewer for the constructive suggestion. The pooled sera samples were used due to sample scarcity. To account for the data accuracy, PRNT assay was done on multiple time points (day 30, day 60 and day 90) for each group. As per reviewer’s suggestion we performed paired students t test between hamster groups immunized with either MR-dCoV or CoV and placebo. Statistical analysis using one-tailed, paired, student’s t test has been provided in the figure 3 and the legends has been updated to include statistical analysis.
Point 6: The data showing no detectable infectious viruses in the lungs of challenged animals (in vaccinated groups) may be somewhat problematic. Based on the significant weight loss after viral challenge in all groups indicating clinical signs in the animals. The fact that no virus was found in the lungs after provocation seems to contradict this observation. I find it difficult to find these data convincing.
Response 6: We acknowledge the comment given by the reviewer. In the challenge studies, we were unable to detect live wild-type challenge viruses (Wuhan or Delta) in the lungs of vaccinated animals collected on day 94 (day 4 post challenge). The observation could be a result of two possible reasons:
- dCoV or MR-COV vaccinated hamsters’ immune system did not allow the wild type challenge virus to replicate. Hence, low or no virus present in lungs could not be detected after passaging in cell line. In support of this hypothesis, live virus could be detected in animals that received either placebo or those vaccinated with MR vaccine.
- The observation that hamsters challenged with wild type virus had transient reduction in weights post challenge can be in part be explained by reduced food intake (supplementary data S5 and S6). Hamsters are very sensitive animals. The intranasal challenge on day 90 involved anesthetization with ketamine and xylazine prior to administration of challenge dose with wild type SARS-CoV2 Wuhan and Delta variants via intranasal route. In support of this hypothesis, transient weight reduction was observed in all animals irrespective of the treatment.
Point 7: Some figures need to be corrected. The legend is cropped, missing important information.
Response 7: We thank the reviewer and acknowledge the comment. The figure has been redone.
Reviewer 2 Report
There are a few typos I will mention just a few of them
SARS-CoV-2is a single stranded positive sense RNA virus, belongingto the family 52
Coronaviridae. SARS-CoV-2 genome expressesfourstructural proteins (spike, membrane, 53
CoV-2.However, many T-cell and B-cell epitopes have been identified within the nucleo- 61
battery of viral antigens are known to induceboth humoral and cellular immunity and 64
The authors did not discuss the efficacy of the vaccine on other strands, it would be nice to discuss the effect of continuous mutations and their relevance to the vaccine
Author Response
Response to Reviewer 2 Comments
Point 1: There are a few typos I will mention just a few of them:
SARS-CoV-2is a single stranded positive sense RNA virus, belongingto the family 52
Coronaviridae. SARS-CoV-2 genome expressesfourstructural proteins (spike, membrane, 53
CoV-2.However, many T-cell and B-cell epitopes have been identified within the nucleo- 61
battery of viral antigens are known to induceboth humoral and cellular immunity and 64
Response 1: We acknowledge the comment given by the reviewer. We noticed that many words were fused possibly due to change in Microsoft office word 2007 to higher version at the time of formatting. The point is well taken accordingly all typos have now been fixed. Revised paper shall be submitted.
Point 2: The authors did not discuss the efficacy of the vaccine on other strands, it would be nice to discuss the effect of continuous mutations and their relevance to the vaccine
Response 2: We thank reviewer for the comment. We assume the reviewer has suggested a discussion on the efficacy of the vaccine on other ‘strains’. We agree with the reviewer that SARS-CoV-2 is constantly mutating with the newly emerging variants have accumulating mutations in the Spike gene. A comment about new variants has been included in the discussion. We would also like to inform that we have conducted a separate study (manuscript under preparation) to study the effectiveness of the dCoV vaccine induced antibodies against SARS-CoV-2 variants. Further, a phase 3 clinical studies in collaboration with WHO (Solidarity Vaccine Trials) is ongoing which is expected to provide real time efficacy data against recently circulating variants.
Reviewer 3 Report
This study may have economical and practical importance by the use of a combination vaccine consisting of Mumps/Rubella and SARS-COVID-2. However, the general scientific impact and interest is not as significant. The scientific approach of this work is similar to the published study in reference 18, concerning the codon pairs de-optimization platform. Additionally, the Mumps/Rubella vaccination approach has been similarly performed before by a number of investigators. Administration of the combined vaccine did not show statistically significant difference in PRINT titers or challenge experiments between monovalent SARS-Covid-2 and the combined vaccine. Text formatting throughout the manuscript and other minor issues are detailed below.
Major: Scientific impact
Minor:
1. Many words are merged in the text throughout the manuscript. Also, spacing before each sentence ought to be addressed.
2. In M&M, no mention of the proportion of SARS-Covid-2 to M/R in the combo vaccine
3. Could the authors explain the reason, I.N. control group was not included. Instead the data appear to rely on a control injected IM
4. Although the number of animals in each group was stated under the figures, the M&M should include this info.
5. The authors considered data in fig. 3 as significant. However, no stats were provided.
6. Fig. 3E (delta) the text of one group is not clear.
7. Ethical concern about use of Hamsters and Biosafety. There is no statement to the effect that the animal experiments has been approved by the biosafety committee and the Committee on Animal Research and Ethics.
Author Response
Response to Reviewer 3 Comments
Point 1: Scientific impact
Response 1: We humbly acknowledge the comment given by the reviewer. The manuscript has huge economic and practical importance. We conducted the earlier study mentioned by the reviewer (Wang et al, 2021) to provide a proof-of-concept of attenuated vaccine strain and its safety when administered via intranasal route. In the present study we showed the candidate vaccine safety when administered via intramuscular route which is more suited for vaccine administration in children. The data presented in this manuscript is pivotal for initiating further clinical studies.
A combination vaccine, a commercial MR with dCoV (MR-dCoV) is intended not to boost the immune response, but to solve a practical problem to the already crowded vaccination schedule in children to avoid multiple needle pricks and thereby enhance vaccination compliance. The study showed that there is no immune interference between the MR and the dCoV vaccine components.
We would also like to add that our study showed for the first time an effectiveness of a live attenuated vaccine against circulating SARS-CoV-2 Delta variant. This is important since live attenuated vaccine, unlike existing Spike protein based vaccines, engage wider immune engagement due to all viral proteins present in the vaccine.
Point 2: Many words are merged in the text throughout the manuscript. Also, spacing before each sentence ought to be addressed.
Response 2: We acknowledge the comment given by the reviewer. We noticed that many words were fused possibly due to change in Microsoft office word 2007 to higher version at the time of formatting. The point is well taken accordingly all typos have now been fixed. Revised paper shall be submitted.
Point 3: In M&M, no mention of the proportion of SARS-Covid-2 to M/R in the combo vaccine
Response 3: We acknowledge the comment given by the reviewer. Material and methods have been updated to include the proportions of dCoV to MR component.
Point 4: Could the authors explain the reason, I.N. control group was not included. Instead the data appear to rely on a control injected IM
Response 4: Placebo group injected by IN route was evaluated in previous studies and no safety concern was observed. Since the end point of the assay was antibody evaluation by PRNT assay, single placebo group was deemed sufficient and in compliance with RRR ethical principles on animal usage.
Point 5: Although the number of animals in each group was stated under the figures, the M&M should include this info.
Response 5: We acknowledge the comment given by the reviewer. Materials and methods section has been updated to include number of animals.
Point 6: The authors considered data in fig. 3 as significant. However, no stats were provided.
Response 6: We acknowledge the comment given by the reviewer. Figure 3 has been updated to include stats.
Point 7: Fig. 3E (delta) the text of one group is not clear.
Response 7: We acknowledge the comment given by the reviewer. Figure 3 has been updated with increased fonts of the text.
Point 8: Ethical concern about use of Hamsters and Biosafety. There is no statement to the effect that the animal experiments has been approved by the biosafety committee and the Committee on Animal Research and Ethics.
Response 8: We acknowledge the comment given by the reviewer. The studies were approved by animal and biosafety committees. The animal ethical statement was made at line No. 207-208. The statement is updated to also include IBC.
Reviewer 4 Report
To whom it may concern,
There appears to be an issue with the presentation of this manuscript. Aside from the remarkably poor English, the paper seems to have many words fused together. The Reviewer couldn't make it past Page 1 (see below):
Line 13-14: “….can pass on passive immunity”---Please re-phrase
Line 15: “Live attenuated virus vaccination approach....”—Please check for grammar.
Line 35: “December2022”….No
Line 36: “Symptomsinclude”…The reviewer suspects that nobody proofread this manuscript
Line 38: “…reportedin childrenpossibly…..”?
The Reviewer stopped reviewing this manuscript at Page 1, due to the excessive number of grammatical/spelling/formatting errors.
Author Response
Response to Reviewer 4 Comments
Point 1: There appears to be an issue with the presentation of this manuscript. Aside from the remarkably poor English, the paper seems to have many words fused together. The Reviewer couldn't make it past Page 1 (see below):
Line 13-14: “….can pass on passive immunity”---Please re-phrase
Line 15: “Live attenuated virus vaccination approach....”—Please check for grammar.
Line 35: “December2022”….No
Line 36: “Symptomsinclude”…The reviewer suspects that nobody proofread this manuscript
Line 38: “…reportedin childrenpossibly…..”?
Response 1: We acknowledge the comment given by the reviewer. Line 13-14 has been rephrased. We noticed that many words (Line 15, 35, 36, 38 and many lines throughout the document) were fused, possibly due change in Microsoft office word version from 2007 (submitted .doc format) to higher version (mdpi format). The point is well taken and all typos have now been fixed. Revised paper has been submitted.
Round 2
Reviewer 1 Report
The authors have addressed my concerns and the manuscript is now acceptable for publication.
Author Response
We sincerely thank the reviewer for taking time to reivew our manuscript and provide constructive suggestions. Minor typos has been fixed.
Reviewer 3 Report
The answers to the reviewer points were not adequately addressed.
Point 1: The practical relevance of this study is OK; however, the scientific impact is low. This is particularly highlighted in Fig. 3E where it is shown there was no statistical significance in the NT on day 90 in the combo vaccine against Measles or Rubella despite an increase in antibody titer.
Point 2: There are still several words fused in the body of the manuscript and in the figure legend.
Point3: No text related to the proportion of SARS-COVId-2 in the combo vaccine is found. No indication which lines in the manuscript.
Point 4: no good justification for not using an IN control. Relying on previous experiments is not appropriate
Point 5: Number of animals in each group was not addressed.
Point 6. It is still unclear which groups are compared for stats. No indication which squares or circles etc..
Author Response
Response to Reviewer's Comments:
Point 1: The practical relevance of this study is OK; however, the scientific impact is low. This is particularly highlighted in Fig. 3E where it is shown there was no statistical significance in the NT on day 90 in the combo vaccine against Measles or Rubella despite an increase in antibody titer.
Response 1: We acknowledge reviewers’ comments. We would like to reinstate that this manuscript highlights the protective efficacy of a live attenuated vaccine against challenge with wild type SARS-CoV-2 Delta variant. Live attenuated vaccine induces a broader immune response compared to the currently available vaccines which are mostly based on single spike protein.
Further, we would like to clarify that in figure 3E, statistical significance was not calculated since these samples were tested on pooled sera samples (pool of 8 sera samples from each group) and was used only as a confirmatory test to evaluate if the antibody levels against Measles and Rubella components are achieved. MR vaccine is a commercial vaccine and its immunogenicity profile has been well studied in humans. We wanted to study the dCoV component which was the principal component under investigation in this study.
Point 2: There are still several words fused in the body of the manuscript and in the figure legend.
Response 2: We agree with the reviewer. Few fused words were found that appeared due to change in version of Ms word. All formatting errors have now been fixed (Line Nos. 295, 299, 323, 344, 365, 404, 600 and 601).
Point 3: No text related to the proportion of SARS-COVId-2 in the combo vaccine is found. No indication which lines in the manuscript.
Response 3: We acknowledge the comment provided by the reviewer. Text related to proportion of SARS-CoV-2 has been provided in the M&M section, line No. 126, in addition to individual virus titers. Virus titers of each vaccine components were provided earlier as virus titers are critical to attenuated vaccines.
Point 4: no good justification for not using an IN control. Relying on previous experiments is not appropriate
Response 4: We agree with the reviewer that relying on previous experiments is not appropriate. We would like to state that the objective of the challenge study was to evaluate immunogenicity and protection conferred by MR-dCoV/dCoV administered via IM route which is the intended route of immunization for children. The safety and immunogenicity profile of commercial MR vaccine via IM route have already been established. The present study aimed at evaluating safety and immunogenicity of dCoV component alone or in combination with MR vaccine components via IM route. dCoV administered via IN route in this study was used only as a comparator for immunogenicity. Hence additional IN placebo group was not part of the experimental design.
Further, we would like to inform the reviewer that we performed parallel study in hamsters wherein dCoV was tested as a booster vaccine both via IM and IN route (Manuscript in preparation). In this study placebo was included for both IM and IN route. No safety concern in the vaccinated or placebo groups were observed in this study. The neutralizing antibody obtained in placebo groups (IM and IN) remained at baseline.
Point 5: Number of animals in each group was not addressed.
Response 5: We acknowledge the comment provided by the reviewer. The number of animals in each group has been included, line nos. 182-183, 189-190 and 209, and elaborated in the result section, line nos. 282-286 and 346.
Point 6. It is still unclear which groups are compared for stats. No indication which squares or circles etc.
Response 6: We acknowledge the comment provided by the reviewer. The figure 3 has been updated for its legends. Text associated with the figures (line no 364, 418) has been updated to clarify the stats.
Round 3
Reviewer 3 Report
The following statement in your reply should be inserted under the "Discussion" section.
"we performed parallel study in hamsters wherein dCoV was tested as a booster vaccine both via IM and IN route (Manuscript in preparation). In this study placebo was included for both IM and IN route. No safety concern in the vaccinated or placebo groups were observed in this study. The neutralizing antibody obtained in placebo groups (IM and IN) remained at baseline"
Author Response
Point 1: The following statement in your reply should be inserted under the "Discussion" section. "We performed parallel study in hamsters wherein dCoV was tested as a booster vaccine both via IM and IN route (Manuscript in preparation). In this study placebo was included for both IM and IN route. No safety concern in the vaccinated or placebo groups was observed in this study. The neutralizing antibody obtained in placebo groups (IM and IN) remained at baseline"
Response 1: We sincerely acknowledge the constructive comment given by the reviewer. We have revised the manuscript and included this comment in the discussion, Line no. 548-552.